# Using contextual factors to elicit placebo and nocebo effects: An online survey of healthcare providers' practice

Leo Druart[1,2]*, Emilie Bailly-Basin[1,3], Maïa Dolgopoloff[2], Giacomo Rossettini[4], Charlotte Blease[5], Cosima Locher[6,7], Alexandre Kubicki[3,8], Nicolas Pinsault[1,2]

1 Univ. Grenoble Alpes, CNRS, VetAgro Sup, Grenoble INP, Grenoble, France, 2 Department of Physiotherapy University Grenoble Alpes, Grenoble, France, 3 Department of Rehabilitation, Franche-Comté University, Montbéliard, France, 4 School of Physiotherapy, University of Verona, Verona, Italy, 5 Department of Psychiatry, Beth Israel Deaconess Medical Center, Digital Psychiatry, Harvard Medical School, Boston, MA, United States of America, 6 Department of Consultation-Liaison Psychiatry and Psychosomatic Medicine, University Hospital Zurich, University of Zurich, Switzerland, 7 Faculty of Health, University of Plymouth, United Kingdom, 8 Laboratoire de neurosciences intégratives, Besançon, France

* leo.druart@univ-grenoble-alpes.fr

**Data Availability Statement:** All relevant data are within the paper and its Supporting Information files.

**Funding:** The author(s) received no specific funding for this work.

## Abstract

Contextual factor use by healthcare professionals has been studied mainly among nurses and physiotherapists. Preliminary results show that healthcare professionals use contextual factors without specifically labelling them as such. The main objective of this study was to evaluate knowledge and explore voluntary contextual factor use among various healthcare professions. The results aim to facilitate hypothesis-generation, to better position further research to explain and characterise contextual factor use. We conducted a web-based questionnaire cross-sectional observational study on a non-probabilistic convenience sample. Face and content validity were tested through cognitive interviews. Data were analysed descriptively. The target population was the main healthcare profession, or final year students, defined by the French public health law. The countries of distribution of the questionnaire were the French-speaking European countries. Among our 1236 participants, use of contextual factors was widespread. Those relating to the therapeutic relationship (e.g., communication) and patient characteristics (e.g., past experiences) were reportedly the most used. Meanwhile, contextual factors related to the healthcare providers' characteristics and their own beliefs were reported as less used. Despite high variability, respondents suggested contextual effects contribute to approximately half of the overall effect in healthcare and were perceived as more effective on children and elderly adults. Conceptual variations that exist in the literature are also present in the way healthcare providers consider contextual effects. Interestingly, there seems to be common ground between how physiotherapists, nurses and physicians use different contextual factors. Finally, in the present study we also observed that while there are similarities across usage, there is lack of both an epistemological and ethical consensus among healthcare providers with respect to contextual factors.

**Competing interests:** The authors have declared that no competing interests exist.

## Introduction

Several reasons can explain treatment improvement. Kleijnen [1] and, more recently, Wampold suggest grouping these reasons into three categories: "natural" effects, specific effects and contextual effects [2]. The so-called "natural" effects are effects that occur spontaneously, due to the dynamics of the condition itself, including the cyclic evolution of symptoms and regression to the mean, without any link to the strategies put in place. These effects are estimated in clinical trials with no-treatment groups [3]. Specific effects are the effects inherently due to a medication or treatment. In the case of medication, they are related to the active pharmacological substance. Clinical trials have been thought out to test these specific effects. They are observed when compared to placebos in randomised clinical trials [3].

Finally, contextual effects are those obtained within the context of the healthcare interaction. This includes behavioural, cognitive and emotional care provided by the therapist [4,5]. Some authors use the term contextual effects as a substitute for placebo effects [4] while others use the term more broadly, including all behavioural, cognitive or emotional care provided [2]. Lastly, an even broader definition exists, including all non-specific effects [6]: i.e. placebo effects, natural history and regression to the mean. This definition is particularly used in studies aiming to determine effect sizes of these categories. Regardless of these, if not simply semantic, conceptual variations, contextual factors (CFs) play a part in patient expectations, the symbolic meaning of a healing setting or the relationship between the healer and the patient [7], influencing non-specific effects by different biological, psychological and social factors.

Although clinical research has aimed to justify treatment use through evaluating specific effects, non-specific effects (i.e., contextual and "natural" effects as defined by Wampold) also contribute significantly to patient improvement [6,8–10]. Although the proportion of contextual effects attributable to placebo effects is still unclear [11,12], Hafliðadóttir et al. showed that the proportion of the treatment effect attributable to context was closely influenced by placebo effects [10]. Interestingly, research has shown that CFs can be used as triggers for placebo and nocebo effects [7,13]. Therefore, we could expect a positive impact on healthcare outcomes if healthcare providers (HCPs) optimise the use of CFs. This implies that HCPs should be aware that placebo effects are part of everyday care, and that CFs lead to maximised placebo effects and minimised nocebo effects [14,15]. Before training HCPs to maximise placebo effects, we need a better understanding of how CFs are currently used across professions. This would allow for a more practice-based education of HCPs, as well as serve as a screening of potential unreasonable use.

However, at the moment it is unclear what HCPs currently know about CFs and, more importantly, if and how they consciously use them in their everyday clinical activities. Initial studies regarding the use of placebo effects or CFs have been conducted in Italy on specialised physiotherapists [16–18], nurses [19] and nursing students [20]. In the Netherlands, a survey focused on nurses and general HCPs [21]. Several studies were also conducted on surgeons both in the United-Kingdom [22,23] and Sweden [24]. These studies, including samples of up to 791 respondents, show that HCPs have some knowledge regarding contextual effects and believe they are effective on healthcare outcomes.

The preliminary results of these surveys indicate that it is likely that HCPs use CFs without specifically labelling them as such. This form of empirical use is forged through clinical practice and through professional know-how learnt before graduating. These surveys mainly focus on specific professions (physiotherapists, nurses, surgeons). However, it is conceivable that distinct health professions perceive the relative importance of CFs differently [25]. Various factors, such as the diverse nature of their activities, the selective processes to access the studies or

even the perception of their discipline's epistemology, could influence HCPs' views of CFs. Therefore, comparisons across different healthcare professions would be of interest. The main objective of this study was to evaluate the knowledge and explore the use of CFs among various healthcare professionals and last year students in France and French-speaking Belgium, and Switzerland. Secondly, our goal was hypothesis-generating, to initiate further research into explaining and characterising CFs use across HCPs.

## Methods

### Study design

To meet this study's objectives, we conducted a cross-sectional observational study on a non-probabilistic convenience sample. Ethical approval was obtained from the local ethics committee for research in the Grenoble-Alpes University (Comité d'Éthique pour la Recherche, Grenoble-Alpes or CERGA) on 07/12/2020 with IRB: CERGA-Avis-2020-2. All participants gave consent to participate. Participants of validity testing gave written consent and participants for the electronic survey were prompted to click "next" to confirm their consent.

### Participants and setting

We surveyed HCPs from European French-speaking countries (France, Switzerland and Belgium). Our participants were required to be currently employed in clinical activities. As there is a broad definition of which professions involved in healthcare are considered HCPs, we based our selection on the French public health law [26].

As a result, our study targeted:

a. HCPs and students in their last year of teaching in the following professions: medical doctors, midwives, dentists, pharmacists, nurses, physiotherapists, occupational therapists, psychomotor therapists, speech therapists, nursing assistants, radiographers, nurse assistants, and orthoptists.

b. Practising or studying in a General Data Protection Regulation (GDPR) compliant country; and

c. Understood the French language.

### Questionnaire development and validity testing

We searched the literature for questionnaires investigating contextual factor use in HCPs that could be adapted in our study and found three [17–19]. However, they were targeted only at physiotherapists or nurses and were therefore not completely suitable for use in this study. Therefore, relying on these questionnaires, we created a more generic, questionnaire suitable for all professions.

To check face and content validity, COSMIN recommendations [27] suggest assessing comprehensiveness and relevance qualitatively with an expert panel. The expert committee was composed of a panel of 4 experts, with both researchers and clinicians (L.D., G.R., A.K. and N.P.), with expertise in the field of placebo studies and/or survey-based research. The panel was solicited both before and after the cognitive interviews described below.

To complete this approach we also ran cognitive interviews [28] through video-calls due to the sanitary restrictions in place at the time. During this step, interviewees were invited to complete the questionnaire while reading and thinking aloud. Meanwhile, the interviewer filled another copy out based on oral justifications given by the participants. Interviewers can

probe the understanding of the questions to test the content validity of the questionnaire. They are a robust way of testing this as we can observe how the survey is handled and the cognitive process behind its completion [29]. Face validity was assessed by observing usability and technical functionality through the screen sharing of the interviewees. One person from each profession was interviewed as well as one student in a medical profession, one in a nonmedical profession and one in a pharmaceutical profession. They were recruited through the professional networks of the authors. Before the interview, an email containing the consent form and information about the study was sent to the participants. The data from the cognitive interviews were anonymized and were not included in the final results of the survey. However, reliability was not tested during the development of this questionnaire.

## Questionnaire description

The questionnaire, available in French (S1 Appendix) and a forward-translated English version (S2 Appendix), was divided into three parts: knowledge of contextual effects, voluntary use of CFs, and socio-demographics.

Participants began with five closed questions about their knowledge on what contextual effects are. First, respondents used 5-point Likert scale (1 = no knowledge to 5 = excellent knowledge) to self-assess their level of knowledge and then the estimated influence these CFs have on their practice. Then, they were asked about the definition, parameters, impact, and mechanisms of contextual effects through closed-ended questions.

In the second part of the questionnaire, participants specify their active use of these effects, and their representations of CFs in their care with 4 closed questions. We first asked to evaluate the perceived importance of several CFs, which all had potential to elicit placebo or nocebo effects, on a linear scale ranging from 0 (no impact on healthcare outcomes) to 100 (fundamental impact on healthcare outcomes). Participants then reported their frequency of intentional use for 12 example CFs identified from literature reviews [4,7] (for example, "Have you ever used titles or status, real or not, to improve the clinical outcome of your care?" followed by the question "how often" if the reply was positive). Respondents were asked about their perceptions of the proportion of the overall effect of care attributable to contextual effects according to patient age, gender, and symptomatology on a scale of 0% to 100%. Finally, we asked participants about their conditions for using CFs. The question was formulated as such: "After having completed the following questionnaire, do you use CFs?" and could be answered "Yes", "No, but I plan to", or "No". To all respondents that didn't answer "No", we asked for their motivations for using CFs. Adaptive questioning reduced the length of this section of the questionnaire. A definition of contextual effects was reminded on pages 3 and 8 of the questionnaire to obtain informed responses and reduce the disparity between participants over lexical discrepancies they could have.

The third part surveys demographic data. Participants provided information on their status, their health disciplines, their ages, and the conditions of their practice.

Respondents were not able to review and change their answers between pages of the survey. Only one question (definition of Contextual Effects) had a randomization of items. Incomplete questionnaires were not registered.

## Recruitment process

This survey was open and self-administered and recruited during two periods of time. The first spread between the 15th of February 2021 to the 1st of April, and the second from the 6th of July 2021 to the 1st of October 2021. LD and EBB distributed the link to the questionnaire by emailing all communication departments of hospitals associated with universities, several

healthcare schools and institutes, and health and social institutions available (public information in France). Communication regarding the study was also conducted on social networks with professional and student associations or unions of various professions. This started a snowballing recruitment process as participants were invited to share the survey.

Due to the recruitment process, participants in this study formed a non-probabilistic convenience sample. This does not allow for the calculation of response rates, nor does it offer generalisations about the wider HCP population. In addition, the process was based on voluntary participation without any incentive.

## Data collection procedure

The questionnaire was encoded on Sphinx Online (www.sphinxonline.com) in conformity with the General Data Protection Regulation of the European Union [30]. When participants click on the link of the questionnaire, an information notice about the survey, data protection, and informed consent appear. Respondents gave their consent to participate by clicking on "next".

Data was anonymous as we collected no cookies, no IP check, no log file analysis, no registration. Data collected were anonymous and non-identifiable. This also meant it was not possible to ensure participants only answered once. All data generated by this research project was stored in compliance with GDPR regulations.

## Statistical analysis

Survey data were downloaded from Sphinx and analysed with R software.

Because we were in an exploratory phase, we collected numerous potential predictors of the use of CFs. The absence of a single outcome of interest has three direct implications for inferential statistical analysis:

First, as the p-value is the probability of getting a test statistic at least as extreme as what was observed if the targeted null hypothesis is true, this last point is mandatory for the statistical test to be relevant. If the null hypothesis of no association is indeed true in the context of randomisation, it cannot be the case in the context of observational data.

Second, as no minimal clinically important difference is stated, no *a priori* sample size has been determined. That is, the power of each predictor test is unknown. This is problematic for both "negative" and "positive" results. In the context of low power, it is well admitted that absence of evidence is not evidence of absence, but it is less known that a significant result is subject to overestimation or direction error (referred respectively as Magnitude and Sign errors by Gelman and Carlin [31]). This means that any result obtained in a context of possible low power is uninterpretable.

Lastly, as every predictor is equally of interest to the authors, every association should be tested, leading to an inflation of the alpha significance level. One possibility would be to adjust for the multiple comparisons, but this does not alleviate the power issue discussed above.

For these reasons, we did not rely on statistical significance to discuss the presence or absence of association. Instead, we discussed graphical representation, whether a pattern emerges and whether the hypothesis is worth testing in future studies.

## Results

We recruited 1236 participants, which were all analysable since incomplete answers were not registered. The median time of completion was 11"49'. A little under half (49.8% n = 616) of our sample accessed our questionnaire through e-mail communications, and 38.6% (n = 477) through social media.

**Table 1. Demographic characteristics of the respondents.**

| Characteristic | Professional N = 995 (80.5%) | Student N = 241 (19.5%) | Total N = 1236 |
|---|---|---|---|
| **Gender**[1] | | | |
| Female | 679 (68%) | 183 (76%) | 862 (70%) |
| Male | 314 (32%) | 56 (23%) | 370 (30%) |
| Other | 2 (0.2%) | 2 (0.8%) | 4 (0.3%) |
| **Age**[2] | 38 (30, 50) | 24 (23, 26) | 34 (26, 47) |
| **Profession**[1] | | | |
| Physiotherapist | 326 (33%) | 74 (31%) | 400 (32%) |
| Nurse | 201 (20%) | 22 (9.1%) | 246 (20%) |
| Physician | 197 (20%) | 49 (20%) | 223 (18%) |
| Other | 100 (10%) | 29 (12%) | 129 (10%) |
| Midwife | 38 (3.8%) | 16 (6.6%) | 54 (4.4%) |
| Speech Therapist | 11 (1.1%) | 27 (11%) | 38 (3.1%) |
| Radiographer | 28 (2.8%) | 8 (3.3%) | 36 (2.9%) |
| Pharmacist | 21 (2.1%) | 11 (4.6%) | 32 (2.6%) |
| Nurse Assistant | 23 (2.3%) | 0 (0%) | 23 (1.9%) |
| Dentist | 19 (1.9%) | 3 (1.2%) | 22 (1.8%) |
| Occupational | 16 (1.6%) | 2 (0.8%) | 18 (1.5%) |
| Surgeon | 8 (0.8%) | Non-Applicable | 8 (0.6%) |
| Psychomotor therapist | 7 (0.7%) | 0 (0%) | 7 (0.6%) |
| **Activity**[1] | | | |
| Private practice | 422 (42%) | | |
| Public sector employee | 391 (39%) | | |
| Private sector employee | 120 (12%) | | |
| Mixed | 55 (5.5%) | | |
| Other | 7 (0.7%) | | |

[1] n (%)

[2] Median (IQR); sorted in descending order of total respondents. Percentages have been rounded up to the tenth of a percent.

## Sample description

Through a period of five months, we recruited a sample of 1236 HCPs, of which 80.5% (n = 995) were professionals, and 19.5% (n = 241) were final-year healthcare students. Among professionals, physiotherapists, nurses and medical doctors were the main professions represented with respectively 33%, 20% and 20%. For students, physiotherapist students, medical students and speech therapists were most represented with respectively 31%, 20% and 11%. The distribution of our population is detailed in Table 1. Among the professionals, private practice, and public employees (42% and 39%, respectively) were the most represented.

## Knowledge regarding contextual effects and contextual factors

Our sample estimated their knowledge of contextual effects to be average (3.08 out of 5 with a SD of 0.89), and that this knowledge had a moderate impact on their clinical practice (3.74 out of 5 with a SD of 0.92). When asked for the definition of contextual effects, we presented our sample with several definitions from the literature: of an inert treatment, the spontaneous course of the disease, a therapeutic encounter, or a placebo/nocebo effect. The two most represented unique choices of our participants were for 67% (n = 833) the definition of placebo or

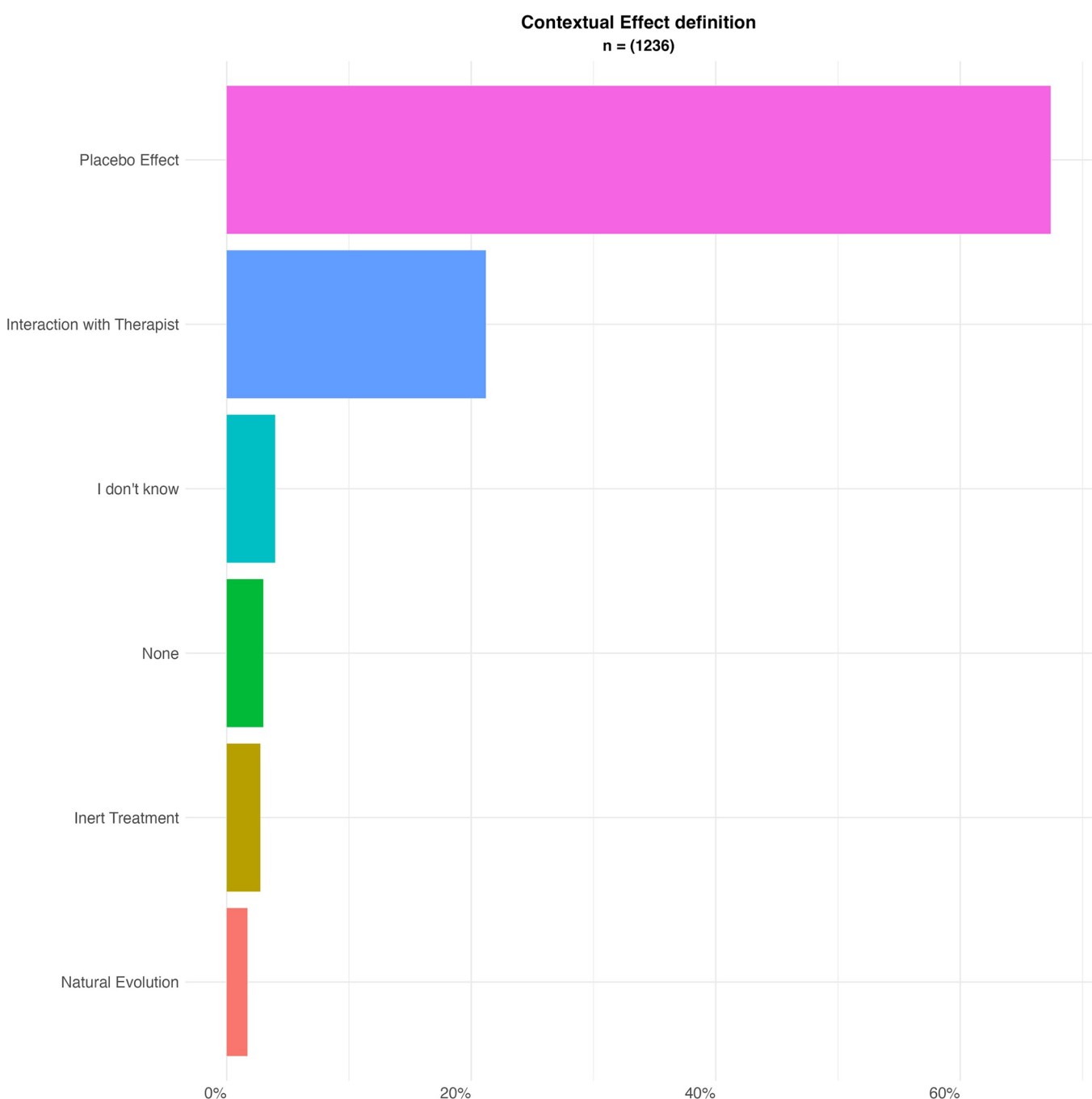

**Fig 1. Definition of contextual effects.**

nocebo effects, and 21% (n = 262) selected the definition of the therapeutic encounter. This is illustrated in Fig 1.

We then asked our sample what influences contextual effects: 95% of the sample agreed that the therapeutic relationship was an influencing factor. The characteristics of the clinical setting, of the therapist and of the patient were influencing factors for 86%, 85% and 82%, respectively. Lastly, the characteristics of the treatment were least consensual as only 68% of our sample thought they influenced contextual effects.

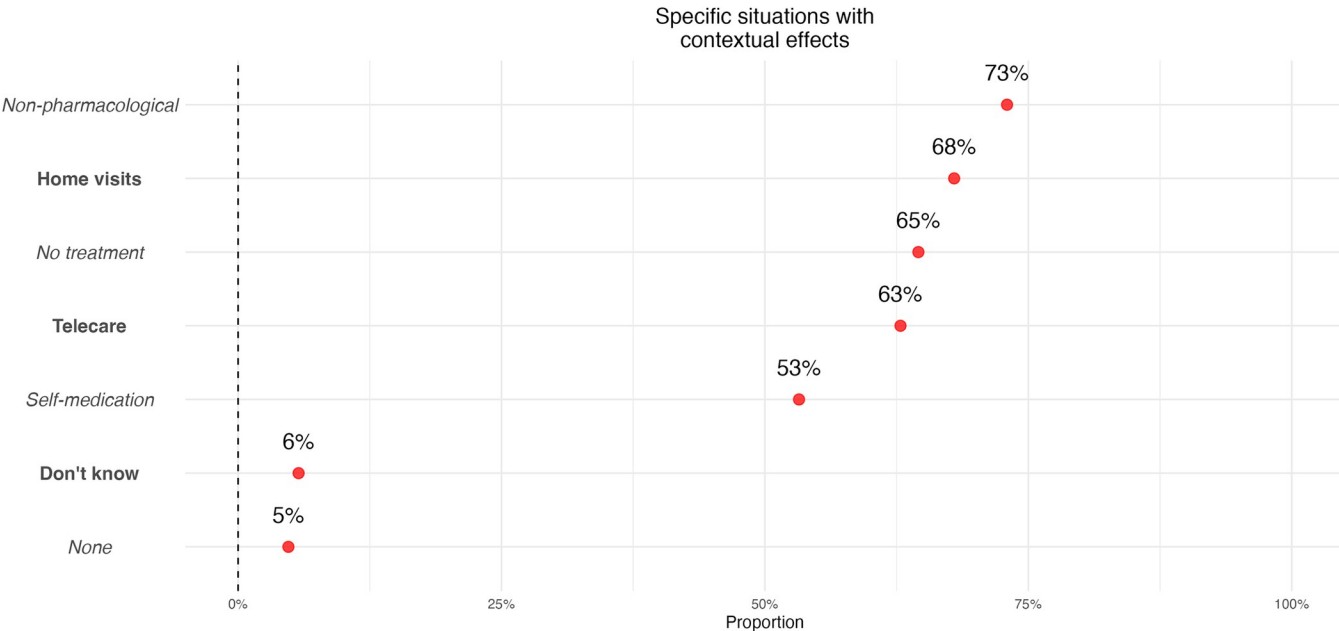

**Fig 2. Specific situations with contextual effects.** Fig 2 (Multiple responses variables, percentage of respondents for each item, n = 1236). Participants were asked whether the following situations were subject to contextual effects: when the treatment was non-pharmacological, in the case of home-visits (i.e. did not take place in a medical setting), when there was no-treatment (i.e. an examination with no prescription), in the case of telecare (i.e. there was no physical presence of a therapist) or when the patients self-medicated (i.e. there was no direct health-encounter) Fig shows the percentage of people who answered "yes".

Several specific situations were then presented where we asked if contextual effects were present. We suggested situations when a non-pharmacological treatment is administered (such as manual therapy), when the consultation takes place at the home of a patient, a home-visit (i.e. the consultation does not take place in a specialised medical environment), when no treatment is delivered during the consultation, when the consultation takes place by means of telecare, when the patients self-medicate (i.e. no HCP is involved in the administration). Although all these situations have the potential to generate contextual effects, of these propositions, only 53% of the panel answered there were contextual effects when the patients self-medicated. The results for the other propositions are presented in Fig 2.

Lastly, we asked about the mechanisms that were responsible for contextual effects. This question allowed for multiple responses and showed that 92% of the sample believed psychological mechanisms were implicated, 81% for suggestions, 67% for conditioning and only 40% for biological mechanisms. For 43% of our sample, these effects were the effect of self-healing processes, and 23% considered them to be due to natural evolution. Lastly, 22% believed other non-identified immaterial entities, such as energies or spirituality, were responsible for these effects. Fig 3 represents these findings.

## Perception of effect size and contextual factor relative importance

Participants were asked to rate on a 100-point scale the weight of several individual CFs to the global contextual effect. The most effective contextual factor, according to our sample, was the therapeutic relationship, followed closely by verbal and non-verbal communication. The CFs related to the patient, such as past experiences and their beliefs and expectations, came next. Physical contact as well as the treatment price were the factors which were perceived as less potent closely followed by the CFs pertaining to the HCP, such as status or therapist expectations. The detailed results for this question are available in the S1 Fig and S1 Table.

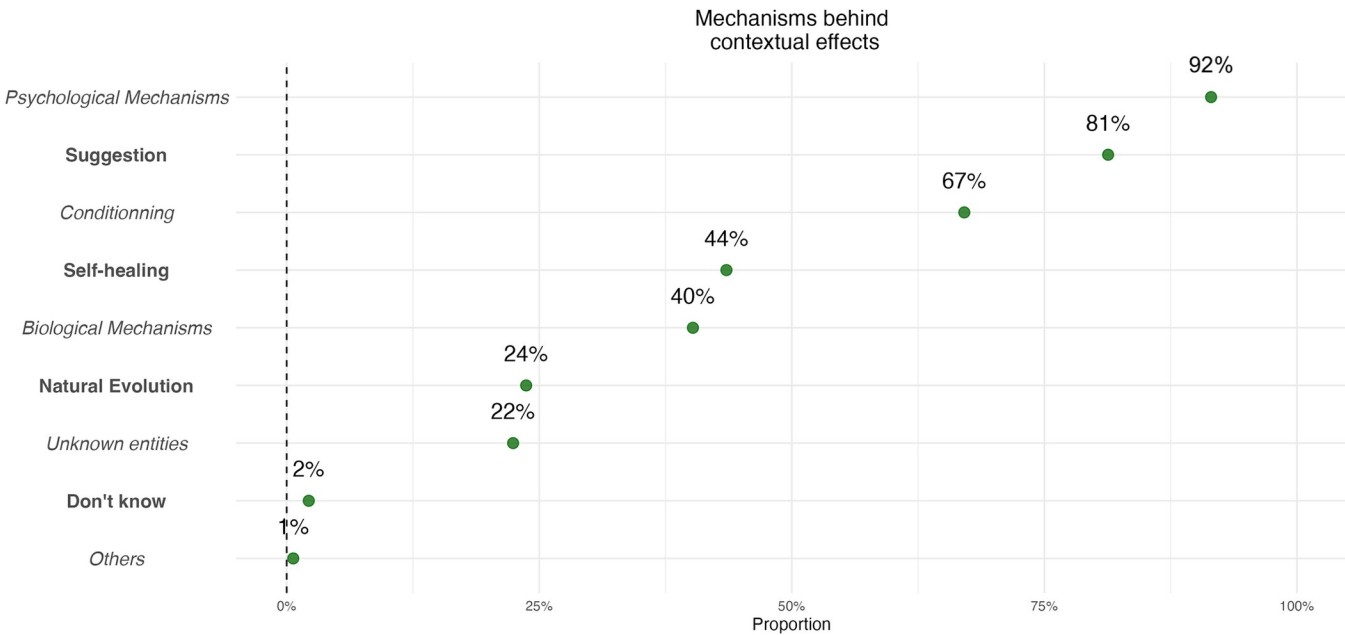

**Fig 3. Mechanisms behind contextual effects.** Fig 3 (multiple responses variables, percentage of respondents for each item, n = 1236).

When asked to estimate the average effect size of CFs, our sample's mean value was 51.5% of the total effect of treatment with a standard deviation of 17.6%. The complete numerical data are available in S2 Table. We then suggested certain situations where the effect size of contextual effects could vary, such as when working with men or women, children, or older adults or when measuring subjective or objective symptoms. Fig 4 shows these results. There seems to be no difference, for our panel, in the effect of CFs in men or women. However, they perceive CFs to work more effectively on younger and older patients compared to average aged patients. There was also a belief that CFs had more of an influence on subjective symptoms rather than objective symptoms. However, these questions were all subject to heavy variability, as seen graphically by the distribution of answers.

## Contextual factor use

When asked if they already voluntarily use CFs in their clinical activity, the large majority (91.7%, n = 1133) of the sample replied that they did, and 5% (n = 67) replied they did not, although they intended to do so in the future. Only 3% (n = 36) replied that they did not use CFs in their clinical activity. Stratifying the respondents according to their evaluation of their knowledge, we plotted the results of CF use. Graphically, it appears that participants who estimated their knowledge lower were less likely to use CFs. This is presented in S2 Fig. Furthermore, when considering the influence of the number of years of practice, it appears that the more experienced HCPs all used CFs as shown in S3 Fig.

The respondents were presented with a list of an example CFs and asked if they used this particular factor. For those replying yes, they were then asked the pace at which they had used this factor. Fig 5 presents the results of this question and S4 Fig details the pace of use. We can see the most used CF is communication, declared by 95% of our HCP sample, followed by patient's past experiences used by 93% of clinicians interviewed. Indeed, the most used CFs are related to either the therapeutic alliance or the patient's characteristics. The least used CFs are

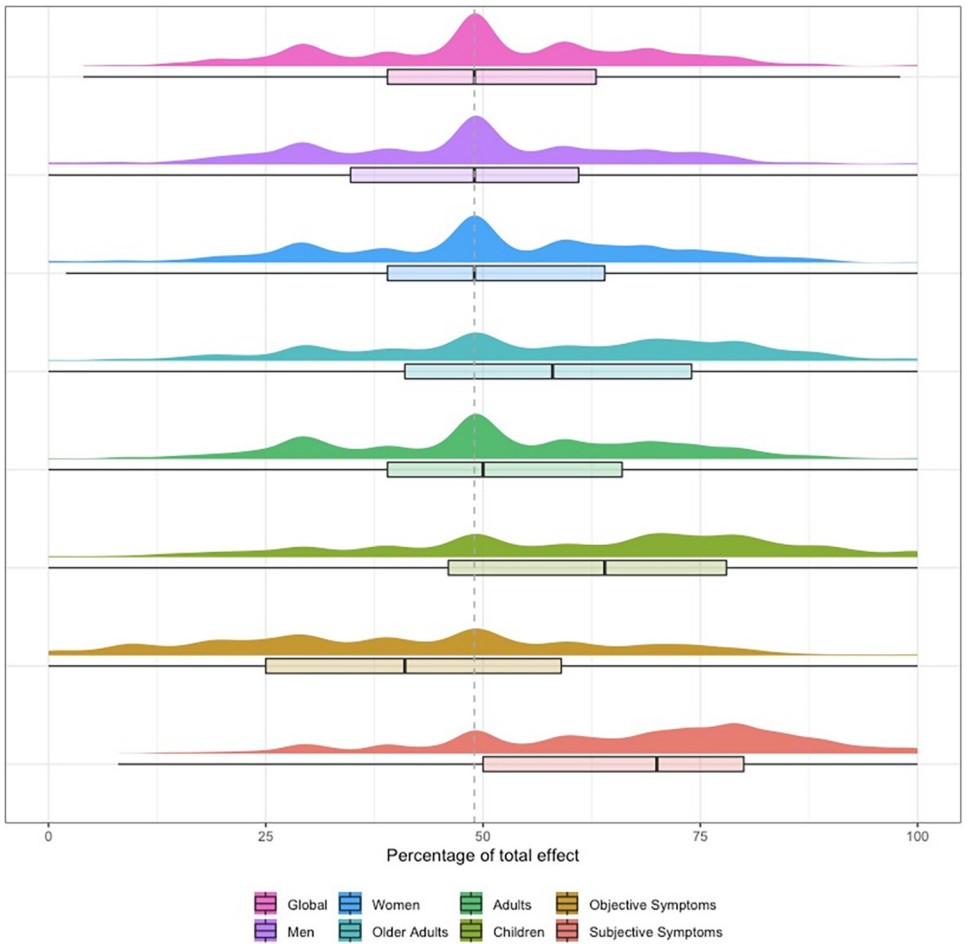

**Fig 4. Perceived proportion of effect attributable to contextual effects depending on patient gender, age, or nature of symptoms.** Fig 4 (n = 1236). From top to bottom: in all conditions, among men, women, older adults, adults, children and on objective and subjective symptoms. Box plots were generated with Q1, Q2 and Q3 quartiles. Distribution is represented by probability density. Dotted line shows Q2 for general population.

those related to the HCP such as a colleague's reputation (52%), own reputation (35%) or one's status (doctor, professor, etc) (31%).

## Healthcare providers' motivation for using contextual factors

The last part of our survey took an interest in the motivations of the HCPs using contextual effects. This question allowed for multiple choices and showed that 83% of the sample actively used CFs to optimise care and 74% to improve patient satisfaction. Some situations were less consensual such as using CFs to limit undesirable effects of a treatment which only 43% declared or using CFs when in a therapeutic impasse which was justified by 32% of the sample. Lastly, 24% of the interviewed HCPs claim to use CFs to compensate for the lack of specific efficacy of a given treatment.

## Intra group comparison

During analysis, as stated in the introduction, we plotted CF use for each profession. Fig 6 shows, for each CF, the use in the three most represented professions in our sample (n>200):

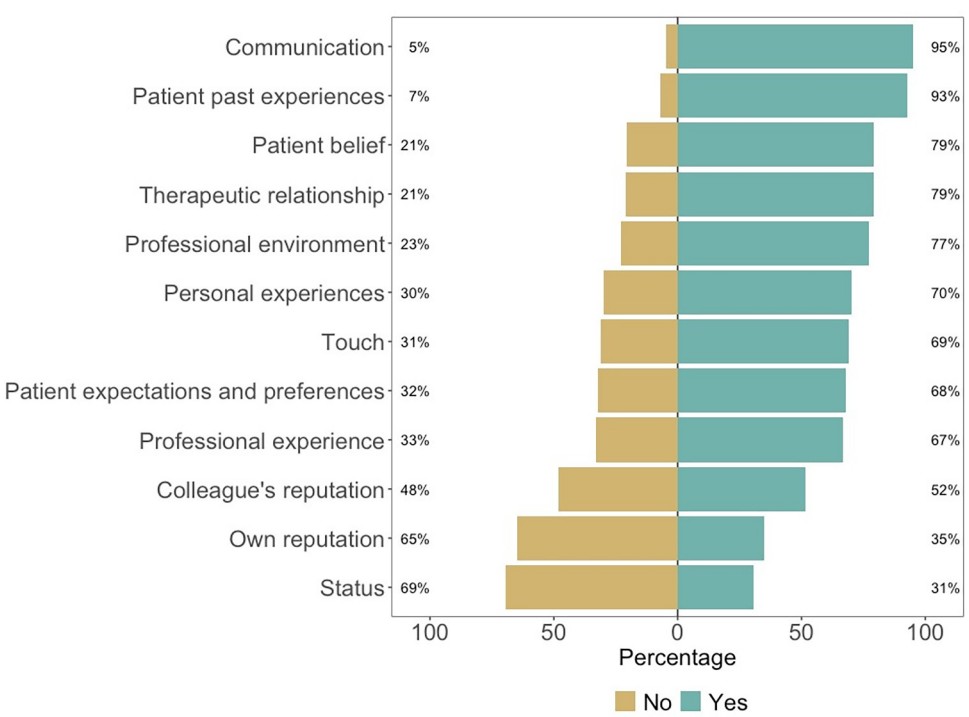

**Fig 5. Contextual factor use.** Fig 5 (n = 1236). Participants answered whether, yes or no, they voluntarily use each contextual factor.

physiotherapists, nurses, and physicians, since the sample sizes are insufficient in other professions. The complete version of this data visualisation is available as S5 Fig. From Fig 6, the use of CFs seems homogenous among physiotherapists, nurses, and physicians. Going further with the comparisons between professions. Comparisons of knowledge and profession were plotted in S6 Fig which suggests there is little to no difference between nurses', physicians' and physiotherapists' perceived knowledge about placebo and nocebo effects.

## Discussion

This study aimed to describe the voluntary use of CFs among healthcare professionals in France and French-speaking Belgium and, Switzerland. Through a web-survey, we led a cross-sectional observational study on a non-probabilistic convenience sample. We gathered 1236 replies, of which all were analysed. From our data, CFs use is widespread. CFs related to the therapeutic relationship (e.g., communication) and the patient (e.g., patients' past experiences or patients' beliefs) are the most used. Meanwhile, CFs regarding the HCP's status or reputation and their own beliefs and past experiences are reported to be less used. Respondents suggested that contextual effects contribute to approximately half of the overall effect in healthcare, although a multimodal distribution showed high variability in responses. Contextual effects were perceived to be more effective on children and elderly adults and were perceived to be similar for men and women. For our participants, subjective symptoms are more susceptible to contextual effects than objective symptoms.

Comparing our results regarding conceptual definitions to previous studies, we notice that, to the best of our knowledge, we are in line with other surveys also showing much diversity in perceptions of definitions [16,18–20,24]. We can find more homogenous answers but only when asking if participants agreed to their suggested definition [22,23]. This is quite different

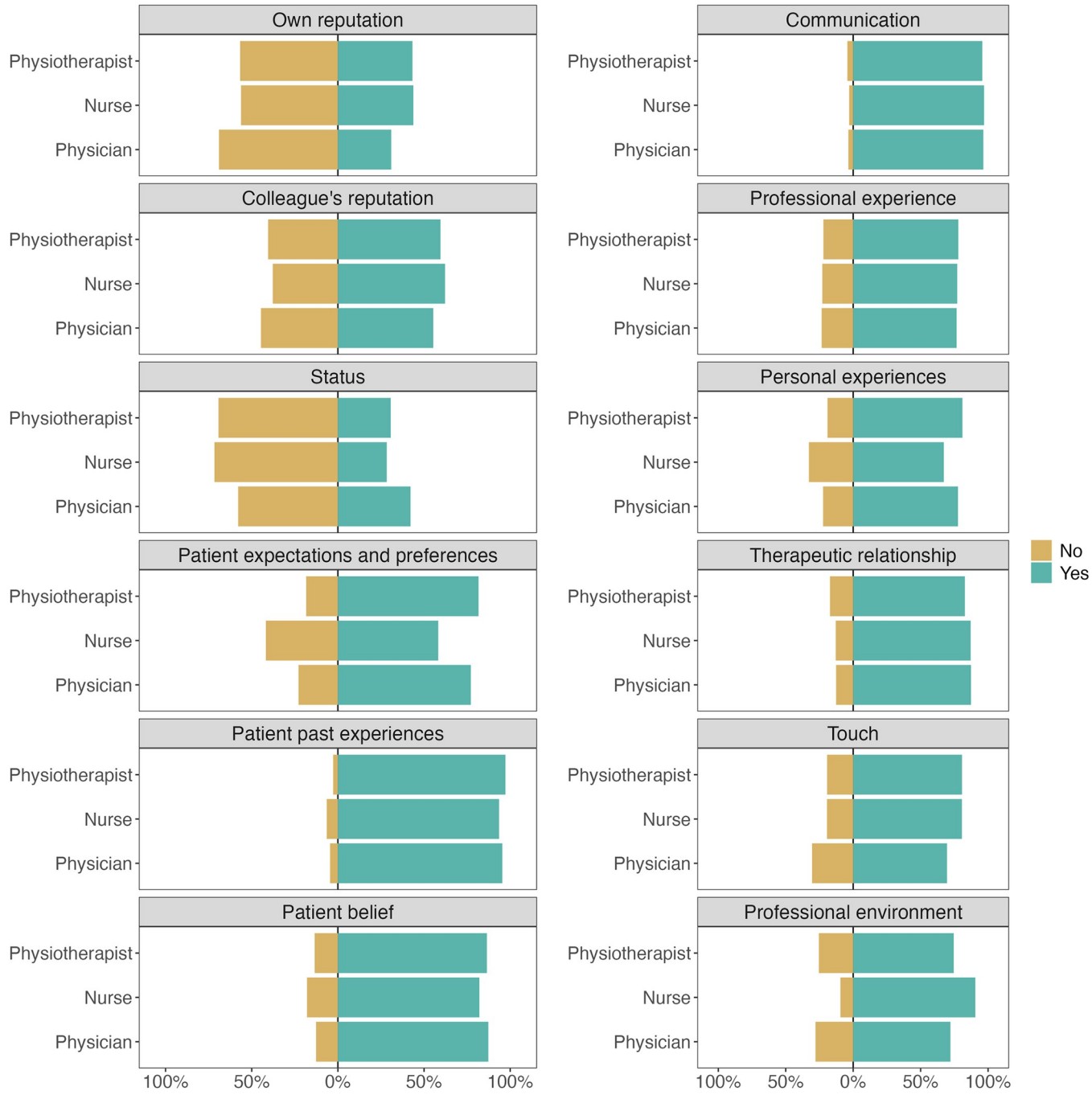

**Fig 6. Use of each contextual factor for physiotherapists (n = 400), nurses (n = 246) and physicians (n = 223).**

from asking to choose a definition among a set number of propositions as done in our study. Thus, the heterogeneity could be due to the fact participants sometimes refer to a broad definition of contextual effects and sometimes restrict their definition to placebo effects. As such, this could reflect the conceptual variations among experts outlined in the introduction section [32,33].

Originally, we also wanted to examine whether participants perceived the omnipresence of contextual effects in care. To this end, we chose to question several situations where contextual

effects were present, but which differed from a usual healthcare encounter. This allowed us to situate our question within a clinical frame to obtain responses less influenced by the question framing. For example, we asked our participants whether they thought contextual effects took place during telecare instead of asking if they existed when the HCP was not physically present. This could have introduced other differentiation factors than those we aimed to investigate (e.g. not the physical presence but the use of technology in our previous example). However, our pre-testing did not lead us to believe this was the case. Interestingly, we can underline inconsistencies between answers on mechanisms and practical implications. As an example, whereas a large majority of participants declared contextual effects were due to psychological mechanisms and conditioning (Fig 2), only 53% considered it was not necessary to meet a HCP for contextual effects to be present (Fig 1). Suggesting that they did not understand those psychological and conditioning mechanisms were not linked to the presence of a HCP. In summary, these original findings show that contextual factors are not well understood: although some mechanisms such as suggestions seem to be identified as mechanisms to elicit placebo and nocebo effects, the way they work does not seem to be well understood.

We found a much higher percentage of CF use (92% for communication) than other surveys (52% for PTs [18] and 53% for nurses [21] for example). We could hypothesise that presenting examples leads to a better illustration of the underlying concepts and thus increases the perception of use. This could be due to using numerous examples of specific CFs in our questions or due to users' over-reporting.

Regarding the proportion of the overall effect attributable to contextual effects, our participants were in line with recent literature [10] although we can reasonably assume that they answered empirically. In fact, considering that the majority (67%) of our sample defined contextual effects as placebo effects, this assumption is quite probable. Similarly to other studies [16,18,19], our panel suggested that the therapeutic alliance was the most impactful CF. Without comparing to other CFs, recent published literature shows that the therapeutic alliance can, in itself, have a clinically significant impact on outcomes [34]. Some studies have highlighted that this view is shared by patients [35]. However, and in contrast to previous findings [16,18–20], our panel showed a belief, unseen-before in the literature, that the least effective CFs were linked to the HCP and the treatment price. This could be interpreted considering cultural specificities of how treatment prices are considered by the social systems in place. Interestingly enough, the most used CFs were those that were perceived as the most effective (linked to the relationship or the patient). However, CFs linked to the therapist were perceived as moderately effective yet were amongst the least used. This suggests that other reasons led our participants not to use them.

These reasons could be linked to the ethics of using CFs in clinical care. Looking closer at the question regarding reasons for the use of CFs, we can see that there is diversity in what some HCPs find acceptable. Although the motivations are quite broad, they show that HCPs seem to find it acceptable to use CFs in everyday clinical work. However, some motivations might be limited in terms of ethical acceptability. For instance, using CFs to compensate for the lack of a specific treatment efficacy seems questionable. A better demonstration of clinically meaningful effects in situations where CFs are optimised needs to be demonstrated. Our results support the need for ethical guidelines regarding the use of CFs preventing unreasonable use, as was previously hinted by expert committees [14,15].

## Implications

Three main implications arise from these findings. Firstly, we can see that the conceptual variations that exist in the literature are also present in the way HCPs consider contextual effects.

Initiatives to find common definitions of CFs are emerging such as a recent consensus study [36]. Secondly, there seems to be common ground on how physiotherapists, nurses and physicians use different CFs. Lastly, we can also see that although there are similarities in usage, there seems to lack both an epistemological (1 of 5 people answered that contextual effects resulted from immaterial entities such as spirits, energies, etc.) as well as an ethical (1 in 4 people saw CFs as a way to justify a treatment otherwise lacking specific effect) common ground. While ethical guidelines are lacking, some general recommendations about how to use CFs exist in the literature. They suggest the use of CFs have relevance in clinical and care settings and should be integrated during the administration of evidence-based treatments to enhance therapeutic outcomes [7]. For example, HCPs should investigate the patient's perspective regarding expectations, beliefs, preferences, mindset and previous experiences, integrating them into the decision-making process [37]. Furthermore, HCPs should optimise therapy administration by being careful with verbal and non-verbal interaction with their patients and the accompanying rituality [37].

## Strengths and limitations

Regarding our study, we can outline a few strengths. Firstly, to the best of our knowledge, our study has the largest sample yet regarding characterisation of CF use. This is mainly due to the recruitment strategy, which had broader inclusion criteria than other studies in the literature since we recruited all professions. Moreover, this is the first study examining the use of CFs in France and, more modestly, other European French-speaking countries. Another originality in this study is to have focused on harnessing placebo and nocebo effects through other means than placebo treatments, whose use is well described in the literature. We focused solely on CFs as enhancers of routine care and not on placebo treatments. This study is also one of the first to have questioned how HCPs perceived the effect size of contextual effects. Although the mean is close to what can be observed in meta-analyses when considering a broad definition of contextual effects, there is an important variance in responses. In some cases, the third quartile reaches up to 80% of the overall effect. These overestimations are not surprising as they are also present for many treatments, as shown, for example, in a survey where 87.7% of general physicians overestimated treatment effects and risks [38]. Lastly, another feature of this questionnaire was its usability for several professions allowing for comparisons between professions. Through pre-testing, we were able to use a questionnaire adapted to multiple professions. We also investigated all categories of CFs through a thorough list.

Even though our study design allowed the strengths mentioned above, it also led to some limitations. Firstly, as our observational study was retrospective, it shares the same bias as other retrospective studies and carries a risk of memory bias from respondents. Secondly, regarding the questionnaire administration, we had no way to determine the number of people who gave up on answering or the total number of people who were exposed to the questionnaire to calculate the participation rate. This could mask a potential selection bias. Although our sampling strategy allowed for a large number of participants, our sample was heterogenous. It had a high proportion of physiotherapists and was constrained in professions such as dentists, surgeons, or nurse assistants. However, the main represented professions are also those who are the most numerous in French HCPs demographics. The same can be observed regarding the geographical localisations of our participants, which are almost exclusively practising in France. Thirdly, regarding the content of our questionnaire, asking about knowledge could have led our participants to have been biased in their responses later. Additionally, we did not check if the responses to questions measuring knowledge were correctly understood. In other surveys, this was done through the use of open-ended questions [21] asking for

examples which could be verified for appropriateness. Similarly, questionnaire reliability was not investigated during questionnaire development.

### Future research

Future research is needed, and the hope is that this exploratory study will inspire follow-up work. Regarding knowledge of contextual effects, qualitative studies (e.g., focus groups or semi-structured interviews) could deepen our knowledge about HCPs' understanding of these effects in routine care and better circumscribe inconsistencies in understanding among HCPs. This could also be completed by qualitative studies looking at how patients perceive effectiveness such as has been done with psychiatric inpatients [35]. Regarding the use of CFs in clinical practice, using the same questionnaire among all professions allows comparable results. Further investigation of CF uses among dentists, nurse assistants, or pharmacists, for example, could be of interest. Similarly, most studies regarding CF use are focused on European countries. More quantitative studies (e.g., surveys) are needed in extra-European countries. This would help better understand if and how cultural determinants could influence HCPs' use of CFs. Furthermore, these studies would only look at the voluntary use of CFs, and qualitative studies are needed on lived experiences of HCPs to better understand their voluntary and involuntary CF use during their clinical reasoning and decision-making process. More diversity could also be sought out by looking at different categories of impairments (e.g., musculoskeletal, neurological, cardio-circulatory, etc.). This could show if some specific types of pathologies are more prone to HCPs using CFs. In line with this, quantifying the declaration bias of such questionnaires would be interesting to see if perceived use matches externally observed use. Finally, researches about CFs and healthcare have to be linked to the discussion about the epistemological foundations that underlie professional practices of each healthcare profession, such as done in psychology [39].

## Supporting information

**S1 Fig. Perceived effect size for individual contextual factors.**
(TIF)

**S2 Fig. Use of contextual factors according to knowledge evaluation.**
(TIF)

**S3 Fig. Use of contextual factors according to professional experience.**
(TIF)

**S4 Fig. Pace of use of individual contextual factors.**
(TIF)

**S5 Fig. Intra-group comparison between different healthcare professions.**
(TIF)

**S6 Fig. Knowledge about contextual factors depending on profession.**
(TIF)

**S1 Table. Numerical values for perceived effect size of individual CFs.**
(DOCX)

**S2 Table. Numerical values for perceived effect size on specific populations.**
(DOCX)

**S1 Appendix. Questionnaire in French.**
(PDF)

**S2 Appendix. Questionnaire in English.**
(PDF)

**S3 Appendix. Questionnaire Logic.**
(PDF)

## Acknowledgments

The authors would like to thank Dr Marco Testa for his advice during the elaboration of the questionnaire.

## Author Contributions

**Conceptualization:** Leo Druart, Emilie Bailly-Basin, Alexandre Kubicki, Nicolas Pinsault.

**Data curation:** Leo Druart, Emilie Bailly-Basin, Maïa Dolgopoloff.

**Formal analysis:** Leo Druart, Maïa Dolgopoloff, Nicolas Pinsault.

**Investigation:** Leo Druart, Emilie Bailly-Basin, Alexandre Kubicki, Nicolas Pinsault.

**Methodology:** Leo Druart, Emilie Bailly-Basin, Maïa Dolgopoloff, Giacomo Rossettini, Alexandre Kubicki, Nicolas Pinsault.

**Project administration:** Leo Druart, Nicolas Pinsault.

**Resources:** Leo Druart, Nicolas Pinsault.

**Software:** Leo Druart, Maïa Dolgopoloff.

**Supervision:** Charlotte Blease, Cosima Locher, Alexandre Kubicki, Nicolas Pinsault.

**Validation:** Giacomo Rossettini, Charlotte Blease, Cosima Locher, Alexandre Kubicki, Nicolas Pinsault.

**Visualization:** Leo Druart, Maïa Dolgopoloff.

**Writing – original draft:** Leo Druart, Emilie Bailly-Basin.

**Writing – review & editing:** Leo Druart, Emilie Bailly-Basin, Maïa Dolgopoloff, Giacomo Rossettini, Charlotte Blease, Cosima Locher, Alexandre Kubicki, Nicolas Pinsault.

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
