## [Decision Letter · Decision Letter 0]

25 May 2023

PONE-D-23-01153

Perception and use of contextual factors in eliciting placebo and nocebo effects: an online survey of healthcare providers in French-speaking countries in Europe

PLOS ONE

Dear Dr. Druart,

Thank you for submitting your manuscript to PLOS ONE. After careful consideration, we feel that it has merit but does not fully meet PLOS ONE’s publication criteria as it currently stands. Therefore, we invite you to submit a revised version of the manuscript that addresses the points raised during the review process.

We look forward to receiving your revised manuscript.

Kind regards,

Elisa Ambrosi

Academic Editor

PLOS ONE

Journal Requirements:

2.Please provide additional details regarding participant consent. In the ethics statement in the Methods and online submission information, please ensure that you have specified what type you obtained (for instance, written or verbal, and if verbal, how it was documented and witnessed). If your study included minors, state whether you obtained consent from parents or guardians. If the need for consent was waived by the ethics committee, please include this information.

Reviewers' comments:

Reviewer's Responses to Questions

**Comments to the Author**

1. Is the manuscript technically sound, and do the data support the conclusions?

Reviewer #1: Yes

Reviewer #2: Yes

2. Has the statistical analysis been performed appropriately and rigorously? 

Reviewer #1: Yes

Reviewer #2: Yes

3. Have the authors made all data underlying the findings in their manuscript fully available?

Reviewer #1: Yes

Reviewer #2: Yes

4. Is the manuscript presented in an intelligible fashion and written in standard English?

Reviewer #1: Yes

Reviewer #2: Yes

5. Review Comments to the Author

Reviewer #1: The article is very interesting and conceptual, but it has some deficits which are given below:

The title is very specific, but there is not much data in the text to emphasize it, and it should be more general.

The main objective in the introduction is not included in the abstract. They gave only the second purpose of the study.

The introduction is too long and you may miss the article's impact while reading. Its flow could be rearranged.

More details on the methodology should be given.

- They used face and content validity methods for tool validation. What is Cohen's Kappa index? And

What is the content validity ratio for each tool item? Was there any question that remained below the critical level and should be removed? How many times did the expert panel?

- Validity tests are recommended for survey development, but reliability testing is necessary. Has a reliability test been done? If so, what is the Chronbach Alpha value? If a pilot study was conducted, how many people was it conducted on? Were the pilot study results included in the total data?

Reviewer #2: In this study, the authors have investigated healthcare provider’s (HCPs) knowledge regarding contextual factors (CF), their use and perception of relative importance. The authors have developed, validated and distributed a web-based questionnaire. They obtained complete answers from 1236 HCPs (995 Professionals + 241 students). This is the largest and most inclusive sample since several professions were represented (physiotherapists, nurses and physicians as well as others) . The results are mostly descriptive and depict a widespread use of CFs amongst HCPs in French speaking Switzerland , with those relating to the therapeutic relationship and to patient characteristics being the most frequent ones. This survey asks timely and helpful questions. Below some comments to further enhance this manuscript.

Major points:

Hypotheses, statistics and results:

It seems the authors could have gone a step further with their data.

-The authors did not make any link between knowledge and use of CFs: are HCPs that have a better knowledge regarding CF more prone to use them in clinic? As you mentioned in the introduction (L113), a better training of HCPs could help to maximize placebo effects… you have data to test if knowledge means application!

-For the intra group comparison, you only compared the use of CFs but not the knowledge and perception of relative effect.

-Is there an impact of the number of years of practice?

Some of the methods/results need clarification:

-You did not clarify: amongst your different propositions of contributing factors what you consider as correct or not.

-L255: did you ensure that each participant only answered once? If not, add to limitations. Were there any duplicates?

-L307: you calculated the average knowledge based on a 5 point Likert scale ranging from 0= no knowledge to 4=excellent knowledge. Yet you reported a mean of 3.08 out of 5. Same comment for the impact of knowledge of the use of CF.

-L351-360: Participants rated on a 100-point scale the weight of CFs but no numerical report has been provided.

-L309: You only reported the most frequently selected definition. A figure could be more relevant to report all the % for each proposed definition.

-L362: a mean (+/- SD) would have been more useful.

-L365: a statistical analysis would have been useful for these comparisons.

-

The discussion could be further enhanced:

-Perhaps summarize some general recommendations for HCPs on how to better use CFs.

Figures and tables that require clarification and improvement:

-Table 1: The order of the items should be consistent (system seems to be highest percentage first). This needs to be improved esp. for gender and profession (random order after “other”; in professions for professionals, there is a total of 100.1%

You reported that 0 orthoptist responded. This information is not relevant

-Figure 1 and 2: please specify what is represented on X axis (% of HCPs ? % of replies?). The marked percentages in the table are too small to read, the graphs could be graphically improved

-Figure 3: What does the grey dotted line correspond to (just below 50% of total effect)?

-Figure 5: all the legends are too small to be read

Minor Comments:

L53-55 : Abstract sentence “ The target population was, or final-year healthcare students as defined by the French public health code.” Needs revision, it makes no sense.

-L456: This sentence is not clear.

-L471: What do you mean by “his” viewpoint?

6. PLOS authors have the option to publish the peer review history of their article (what does this mean?). If published, this will include your full peer review and any attached files.

Reviewer #1: No

Reviewer #2: No

---

## [Author Response · Author response to Decision Letter 0]

12 Jul 2023

Dear Editor and reviewers,

We would like to thank the Editor and the reviewers of PLOS One for their interest in our manuscript (Manuscript ID: PONE-D-23-01153: “Perception and use of contextual factors in eliciting placebo and nocebo effects: an online survey of healthcare providers in French-speaking countries in Europe”). We have carefully considered your valuable comments that improved the manuscript's overall quality, considering the criticism and feedback of experts in the field. As you can see, we have changed the manuscript to improve the manuscript. We hope to have addressed all concerns. Please find below our point-by-point responses to the peer reviewer's comments. We have highlighted each change in the manuscript using tracked changes and included these changes to the reply to the reviewers with references to the line numbering in the tracked changes manuscript. 

Thank you again for giving us the possibility to revise the manuscript in-depth for your full consideration.

Regards,

Leo Druart

Reviewer #1 comments

The article is very interesting and conceptual, but it has some deficits which are given below:

1. The title is very specific, but there is not much data in the text to emphasize it, and it should be more general.

The title was changed from “Perception and use of contextual factors in eliciting placebo and nocebo effects: an online survey of healthcare providers in French-speaking countries in Europe” to “Using contextual factors to elicit placebo and nocebo effects: an online survey of healthcare providers’ practice”

2. The main objective in the introduction is not included in the abstract. They gave only the second purpose of the study.

We have modified the abstract to be clearer on the aims of this study. The abstract has been rephrased as such on lines 26-28:

“The main objective of this study was to evaluate knowledge and explore voluntary contextual factor use among various healthcare professions. The results aim to allow hypothesis-generating initiating further research explaining and characterising contextual factor use.”

3. The introduction is too long and you may miss the article's impact while reading. Its flow could be rearranged.

The introduction has been streamlined to better showcase the impact of this article. As such, several passages have been either deleted or reformulated. For example, the following passage has been deleted: “As an example, a recent meta-analysis showed that, across all conditions, half (0.54, 95%CI 0.46 to 0.64) of the overall treatments’ effects could be attributed to contextual effects [10]. For osteoarthritis, this proportion was closer to 75% (0.75, 95%CI 0.24 to 0.68) [6] and for fibromyalgia around 60% (0.60, 95%CI 0.56 to 0.64) [8]. These high proportions justify the need to understand them better.” As another example, the following passage has also been deleted: “The percentage of HCPs that deliberately used CFs was estimated to be 52% for physiotherapists [18] and 53% among nurses [21]. When asked what CFs they believed to be most effective in generating placebo or nocebo effects, respondents put forth the therapeutic relationship and patient expectations in Bisconti et al. [16]. Whereas in Rossettini et al.‘s sample [18] they added the patient-centred approach Whether it was nurses or physiotherapists, the factors believed to be less effective were those linked to the HCP [18,19].” These changes should allow to provide the readers with a clear understanding of the concepts discussed in this article followed by the relevance of the topic, the state of the current knowledge on the topic and concludes with the objective of this study.

More details on the methodology should be given.

4. They used face and content validity methods for tool validation. What is Cohen's Kappa index? And What is the content validity ratio for each tool item? Was there any question that remained below the critical level and should be removed? How many times did the expert panel?

To assess face and content validity, we have followed the COSMIN recommendations (Measurement in Medicine, de Vet et al. p154-159) who suggest one way of measuring face and content validity is subjectively. This requires reflecting on the survey’s relevance and comprehensiveness. This approach was completed with cognitive interviews which also rendered qualitative results. Consequently, we do not have any quantitative outcome measures for the validity of our survey such as Cohen’s Kappa or Alpha’s Cronback or Gwet’s AC. To make sure our methods are clearer, we went into more detail when describing this and added reference to methodological guidelines followed. As such, this results in the following addition to the manuscript on lines 153-156: “To check face and content validity, COSMIN recommendations [27] suggest assessing comprehensiveness and relevance qualitatively with an expert panel. The expert committee was composed of a panel of 4 experts, with both researchers and clinicians (L.D., G.R., A.K. and N.P.), with expertise in the field of placebo studies and/or survey-based research. The panel was solicited both before and after the cognitive interviews described below.”

5. Validity tests are recommended for survey development, but reliability testing is necessary. Has a reliability test been done? If so, what is the Chronbach Alpha value? 

Reliability was not assessed in this study. Indeed, the International Handbook of Survey Methodology* has suggested that there is little need to proceed with in-depth investigations for validity or reliability for surveys. Therefore, we did not investigate both these parameters. Instead, they insist on the need of investigating face validity among experts and including a pre-test phase in a sample of the target population followed by debriefing sessions which we have performed.

However, recognizing this is a limitation to our study and to ensure this is clear to the readers we have explicitly mentioned this in the methods and added this to the limitations. As such, “However, reliability was not tested during the development of this questionnaire.” was added on lines 166-167 and “Similarly, questionnaire reliability was not investigated during questionnaire development.” was added on lines 459-460. We hope this may address any concern of the reviewer.

*D. de Leeuw , Joop Hox, Don Dillman. International Handbook of Survey Methodology (European Association of Methodology Series). Taylor and Francis group. 1st ed. New York (USA).

6. If a pilot study was conducted, how many people was it conducted on? Were the pilot study results included in the total data?

No pilot study has been carried out in addition to cognitive interviews. The data from the cognitive interviews was not included into the database. This has been made explicit in the manuscript of lines XX: “and were not included in the final results of the survey.”

 

Reviewer #2 comments

In this study, the authors have investigated healthcare provider’s (HCPs) knowledge regarding contextual factors (CF), their use and perception of relative importance. The authors have developed, validated and distributed a web-based questionnaire. They obtained complete answers from 1236 HCPs (995 Professionals + 241 students). This is the largest and most inclusive sample since several professions were represented (physiotherapists, nurses and physicians as well as others). The results are mostly descriptive and depict a widespread use of CFs amongst HCPs in French speaking Switzerland, with those relating to the therapeutic relationship and to patient characteristics being the most frequent ones. This survey asks timely and helpful questions. Below some comments to further enhance this manuscript.

Major points:

Hypotheses, statistics and results:

It seems the authors could have gone a step further with their data.

7. The authors did not make any link between knowledge and use of CFs: are HCPs that have a better knowledge regarding CF more prone to use them in clinic? As you mentioned in the introduction (L113), a better training of HCPs could help to maximize placebo effects… you have data to test if knowledge means application!

Fear about overloading our results, we restrained from going further believing it was better to maintain a focus. Following this reviewer’s advice, however, we will now add these comparisons. We plotted the perception of knowledge to usage and added this figure to the supplementary materials. We also added the following description to the manuscript on lines 316-320: “Stratifying the respondents according to their evaluation of their knowledge, we plotted the results of CF use. Graphically, it appears that participants who estimated their knowledge lower were less likely to use CFs. This is presented in S7 Fig.”

8. For the intra group comparison, you only compared the use of CFs but not the knowledge and perception of relative effect.

This has been plotted in a figure which we added to the supplementary materials to not overload the article itself. The following comment was incorporated into the manuscript to detail this: “Going further with the comparisons between professions. Comparisons of knowledge and profession were plotted in S11 Fig which suggests there is little to no difference between nurses’, physicians’ and physiotherapists’ perceived knowledge about placebo and nocebo effects.”

9. Is there an impact of the number of years of practice?

The number of years of practice may lead to more use of recognition of CFs. We have added this in S8 Fig along with the following text on lines 

318-320: “Furthermore, when considering the influence of the number of years of practice, it appears that the more experienced HCPs all used CFs as shown in S8 Fig.”

Some of the methods/results need clarification:

10. You did not clarify: amongst your different propositions of contributing factors what you consider as correct or not.

We added this information in the description of the questionnaire subsection on lines 184-185 as follows: “which all had potential to elicit placebo or nocebo effects”

11. L255: did you ensure that each participant only answered once? If not, add to limitations. Were there any duplicates?

As we wished to not collect any identifying data, we did not collect any IP addresses to ensure no one could reply twice. However, we checked for duplicate responses and found none. We have now explicitly added this to the data collection subsection section of the manuscript: “This also meant it was not possible to ensure participants only answered once.”

12. L307: you calculated the average knowledge based on a 5-point Likert scale ranging from 0= no knowledge to 4=excellent knowledge. Yet you reported a mean of 3.08 out of 5. Same comment for the impact of knowledge of the use of CF.

The values were ranged from 1 to 5. To make sure there is no further confusion possible for future readers, we have made this more explicit on lines 180 of the manuscript.

13. L351-360: Participants rated on a 100-point scale the weight of CFs but no numerical report has been provided.

We have added the numerical data providing means and SDs in the supplementary materials (S5) to not overload the results section.

Importance of individual contextual factors Mean (%) SD

Quality of the therapeutic relationship 87.2 13.8

Verbal and non-verbal communication 83.5 16.8

Patient beliefs and representations on symptoms 83.0 16.4

Patient expectations and preferences 81.0 17.3

Patient past-experiences 78.7 19.0

Professional status 74.4 19.7

Professional reputation 74.2 21.1

Healthcare setting 72.8 20.1

Therapist beliefs and representations 70.0 22.6

Therapist previous experiences 66.9 23.2

Physical contact with patient 60.7 24.9

Treatment price 54.6 25.3

14. L309: You only reported the most frequently selected definition. A figure could be more relevant to report all the % for each proposed definition.

We have added a figure to better illustrate this in the manuscript. This is now Fig 1 and has been added on lines 258-260.

15. L362: a mean (+/- SD) would have been more useful.

This has been updated in the manuscript of lines 300-301: “When asked to estimate the average effect size of CFs, our sample’s mean value was 51.5% of the total effect of treatment with a standard deviation of 17.6%.” We have also added the numerical values to the supplementary materials in the following table.

Proportion of effect attributable to contextual factors Mean (%) SD

Overall 51.5 17.6

In women 51.9 18.7

In men 49.3 18.7

In children 61.3 22.3

In adults 52.4 18.7

In older adults 57.6 21.0

For subjective symptoms 66.9 19.1

For objective symptoms 43.0 21.9

16. L365: a statistical analysis would have been useful for these comparisons.

We have detailed in lines 218 to 236 why we did not use association testing in this analysis. For this reason, we did not add any significance testing. However, to avoid any confusion for the readers, we have formulated our interpretations with more caution towards causal language. 

The discussion could be further enhanced:

17. Perhaps summarize some general recommendations for HCPs on how to better use CFs.

We have added this to the implications of the study. The following paragraph has been added: “While ethical guidelines are lacking, some general recommendations about how to use CFs exist in the literature. They suggest the use of CFs have relevance in clinical and care settings and should be integrated during the administration of evidence-based treatments to enhance therapeutic outcomes [7]. For example, HCPs should investigate the patient's perspective regarding expectations, beliefs, preferences, mindset and previous experiences, integrating them into the decision-making process [36]. Furthermore, HCPs should optimise therapy administration by being careful with verbal and non-verbal interaction with their patients and the accompanying rituality [36].”

Figures and tables that require clarification and improvement:

18. Table 1: The order of the items should be consistent (system seems to be highest percentage first). This needs to be improved esp. for gender and profession (random order after “other”; in professions for professionals, there is a total of 100.1%

You reported that 0 orthoptist responded. This information is not relevant.

The sorting order has been rearranged to be follow the following logic: in descending order for total column. This has been explicated in the table’s legend. Gender has been rearranged to fit the same sorting rule. The total adding up to 100.1% is due to rounding in up to the tenth of a percent, this has also been added to the legend of the table. 

Orthoptists were reported to show that we aimed to recruit some but were unsuccessful. Following the reviewer’s suggestion, we have deleted this. 

Table 1 is now updated following this reviewer’s comments, we believe readability of the data is cleared now.

19. Figure 1 and 2: please specify what is represented on X axis (% of HCPs? % of replies?). The marked percentages in the table are too small to read, the graphs could be graphically improved.

We have added this to the legend of the figure: the X axis shows % of respondents with the n specified in the title of the figure. The graphs have been reworked to improve readability. We hope the changes will be satisfactory to this reviewer.

20. Figure 3: What does the grey dotted line correspond to (just below 50% of total effect)?

The grey dotted line shows the mean for the first graph the global effect. This line helps in showing whether participants considered the population to be more or less responsive to contextual factors compared to the general population. This was indeed forgotten in the legend. Please excuse this oversight which we have now corrected.

21. Figure 5: all the legends are too small to be read

We have increased label size to improve visibility.

Minor Comments:

22. L53-55: Abstract sentence “The target population was, or final-year healthcare students as defined by the French public health code.” Needs revision, it makes no sense.

The sentence has been revised as part of the sentence had been deleted in a previous revision of the manuscript. It now reads as “The target population was the main healthcare profession, or final year students, defined by the French public health law.”

23. L456: This sentence is not clear.

This sentence has been rephrased as follows on lines 389-391: “In summary, these original findings show that contextual factors are not well understood: although some mechanisms such as suggestions seem to be identified as mechanisms to elicit placebo and nocebo effects, the way they work does not seem to be well understood.”

24. L471: What do you mean by “his” viewpoint?

This was a typo and was meant to write “this viewpoint.” It has been corrected.

---

## [Decision Letter · Decision Letter 1]

22 Aug 2023

Using contextual factors to elicit placebo and nocebo effects: an online survey of healthcare providers’ practice

PONE-D-23-01153R1

Dear Dr. Druart,

We’re pleased to inform you that your manuscript has been judged scientifically suitable for publication and will be formally accepted for publication once it meets all outstanding technical requirements.

Kind regards,

Elisa Ambrosi

Academic Editor

PLOS ONE

Additional Editor Comments (optional):

Reviewers' comments:

Reviewer's Responses to Questions

**Comments to the Author**

1. If the authors have adequately addressed your comments raised in a previous round of review and you feel that this manuscript is now acceptable for publication, you may indicate that here to bypass the “Comments to the Author” section, enter your conflict of interest statement in the “Confidential to Editor” section, and submit your "Accept" recommendation.

Reviewer #2: All comments have been addressed

2. Is the manuscript technically sound, and do the data support the conclusions?

Reviewer #2: Yes

3. Has the statistical analysis been performed appropriately and rigorously? 

Reviewer #2: Yes

4. Have the authors made all data underlying the findings in their manuscript fully available?

Reviewer #2: Yes

5. Is the manuscript presented in an intelligible fashion and written in standard English?

Reviewer #2: Yes

6. Review Comments to the Author

Reviewer #2: (No Response)

7. PLOS authors have the option to publish the peer review history of their article (what does this mean?). If published, this will include your full peer review and any attached files.

Reviewer #2: No

---

## [Editor Report · Acceptance letter]

25 Aug 2023

PONE-D-23-01153R1 

Using contextual factors to elicit placebo and nocebo effects: an online survey of healthcare providers’ practice 

Dear Dr. Druart:

I'm pleased to inform you that your manuscript has been deemed suitable for publication in PLOS ONE. Congratulations! Your manuscript is now with our production department. 

Kind regards, 

on behalf of

Dr. Elisa Ambrosi 

Academic Editor

PLOS ONE